# PR-VTON: Enhancing Detail Fidelity in Virtual Try-On with Refined Positional Encoding

| Any2AnyTryon | FitDiT | Leffa | IDM-VTON | Ours |
|---|---|---|---|---|

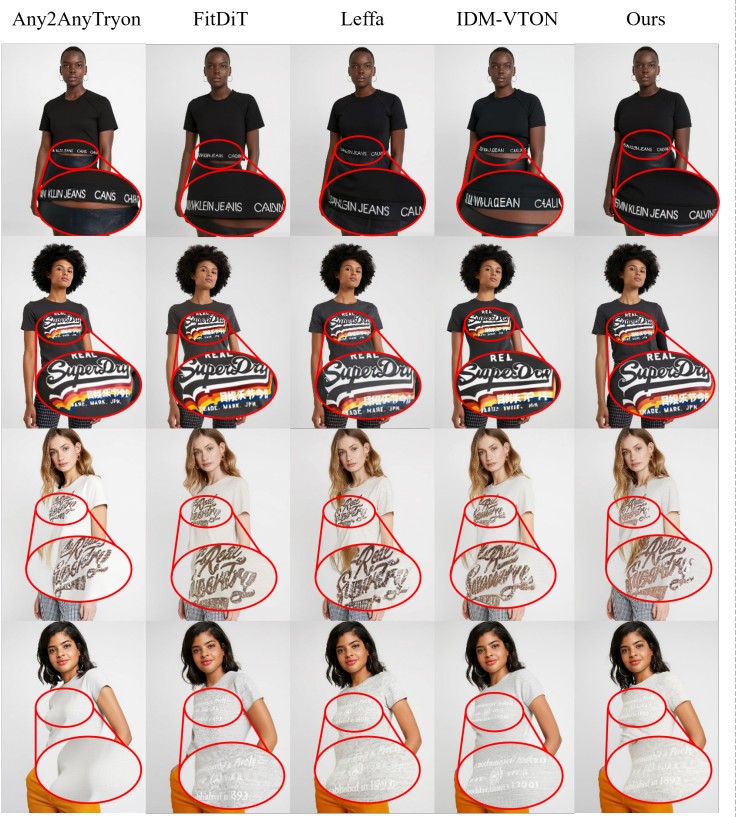 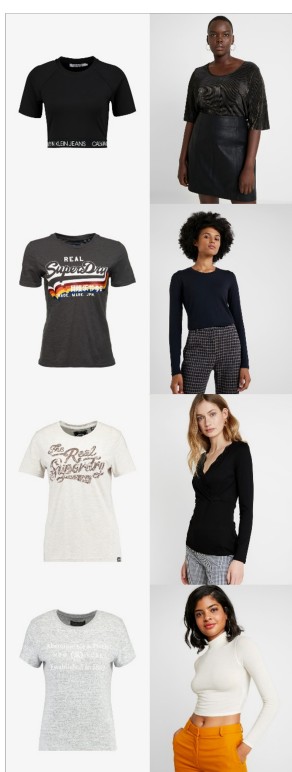

Figure 1: State-of-the-art virtual try-on models still struggle to render high-resolution fine details. Our PR-VTON refines positional encodings at the PE level to steer attention and feature learning, markedly improving fine-grained texture preservation in the generated try-on results.

## ABSTRACT

Recent advancements in pre-trained diffusion models have significantly enhanced image-based virtual try-on, enabling the realistic synthesis of garments for simple textures. However, preserving high-frequency patterns and text consistency remains a formidable challenge, as existing methods often fail to retain fine-grained details. To address this, we introduce PR-VTON, a simple yet effective method that integrates a Position-Refined Positional Encoding (termed PRPE) and a lightweight positional relation learning module (termed PRL) to enhance detail preservation across diverse fabric designs. Specifically, PRPE leverages the inherent impact of positional encoding on attention mechanisms within the Diffusion Transformer (DiT) architecture, guiding attention maps with precise positional cues to achieve superior texture fidelity without additional modules or complex loss functions. Meanwhile, PRL explicitly models token-level correspondences between garments and target bodies, ensuring accurate spatial alignments. Extensive experiments on standard benchmarks demonstrate that PR-DIT surpasses

existing baselines in both quantitative and qualitative metrics, with marked improvements in perceptually sensitive areas, such as textual logos. Furthermore, we critically reassess evaluation protocols for virtual try-on, highlighting deficiencies in existing metrics for capturing global consistency and fine detail fidelity,and propose a detail-focused metric loc-CMMD, establishing a more robust standard for high-resolution virtual try-on research.

# 1 INTRODUCTION

Virtual try-on (VTON) has become a pivotal task, propelled by the rapid expansion of e-commerce. Given a reference garment image and a target person image, VTON aims to generate a realistic try-on outcome, seamlessly aligning the garment with the individual's body shape while preserving intricate, fine-grained textures. This technology significantly enhances user experience and delivers accurate product visualizations, ultimately contributing to reduced return rates.

Before the diffusion era, VTON methods primarily relied on appearance flow estimation coupled with GAN-based synthesis Han et al. (2018); Wang et al. (2018); Yang et al. (2020); Ge et al. (2021); Gou et al. (2023); Xie et al. (2023); Yang et al. (2024). However, the limited generative capacity of GANs Goodfellow et al. (2020) frequently resulted in severe artifacts and unrealistic texture distortions. The emergence of diffusion models prompted a reversal of this situation. Current approaches typically inject garment features into the generative process through reference networks or in-context learning, leveraging powerful pre-trained diffusion models to achieve globally coherent results Zhu et al. (2023); Morelli et al. (2023); Kim et al. (2024); Zeng et al. (2024); Choi et al. (2024); Chong et al. (2024); Zhou et al. (2025). Recent efforts have further pursued fine-grained control by constraining garment regions or optimizing feature injection. Despite these advancements, even state-of-the-art models built on Diffusion Transformer (DiT) Peebles & Xie (2023) backbones, such as Stable Diffusion 3 Esser et al. (2024) or Flux, struggle to preserve high-frequency details faithfully. In particular, small textual logos and intricate textures are often lost or distorted, as shown in Figure 1.

Unlike previous methods that establish semantic positional correspondences through feature injection and attention constraints, we leverage the inherent properties of positional encodings (PEs) within Diffusion Transformer (DiT) architectures to naturally represent spatial relationships. By refining PEs at the positional level, rather than the feature level, we impose precise constraints based on positional relationships. As PEs influence feature interactions through the attention mechanism, subtle adjustments at the PE level effectively guide feature interactions, significantly enhancing texture detail, consistency, and overall garment fit. Furthermore, since PE refinement operates independently of the model, it requires neither complex architectural modifications nor bespoke loss functions terms to constrain features, markedly improving usability and scalability. The technique integrates seamlessly with any DiT backbone and is readily extensible toward unified editing frameworks.

To illustrate how positional relations influence feature interactions, we conducted an in-context learning (ICL) experiment, as depicted in Figure 2. Using an inpainting model not specifically trained for virtual try-on, we concatenate a masked image with a reference garment image (cloth). The experiments demonstrate that treating the concatenated image as a single entity with default positional encoding (Fig. 2(1)) results in a generated image that fails to incorporate the input garment's details entirely, as the untrained model cannot accurately reference clothing features. To address this, we attempt to guide the model to attend to the reference image by using positional encoding. In Fig. 2(2), we paste the positional coordinates of the masked image onto the reference image for encoding, inducing a coordinate-aligned copy effect, even without training, where the garment is essentially copied into the masked region with remarkably preserved details, despite incorrect positional alignment. Based on this, we hypothesize that incorporating precise positional correspondences could enable the model to generate spatially accurate and detail-rich results. Consequently, in Fig. 2(3), We manually define a coordinate mapping $\phi$ that assigns identical coordinates to semantically corresponding points on the garment and the target person. This yields results where both fine details and spatial relations are effectively aligned, demonstrating the strong semantic alignment capability of positional mapping and its potential to preserve high-fidelity details from the reference image.

Motivated by this zero-shot behavior, we introduce a Position-Refined Positional Encoding (PRPE): instead of relying on fixed, image-agnostic PEs, we reconstruct and remap positional encodings according to estimated garment-to-body correspondences. By injecting these alignment-aware PEs into the DiT attention stack, PRPE implicitly steers attention to aggregate reference textures along semantically correct paths, improving both global plausibility and high-frequency detail fidelity—without introducing extra modules or bespoke loss terms on the diffusion features. To make the approach broadly applicable, we further present a lightweight Positional Relation Learning (PRL) module that learns a stable remapping field directly from in-context pairs, obviating any pre-computed matches or labels. Beyond methodology, we revisit evaluation for VTON. We find that widely used metrics fail to reliably capture global consistency and fine detail at high resolution: a naïve warped-paste baseline can score near state-of-the-art despite poor perceptual quality, underscoring a gap with human judgment. To address this, we adopt stronger perceptual backbones (CMMD Jayasumana et al. (2024), FD-DINOv2 Stein et al. (2023); Ahn (2024), DreamSim Fu et al. (2023); Sundaram et al. (2024)) that better correlate with human assessments, and we introduce loc-CMMD, a metric tailored to textual and micro-texture fidelity. loc-CMMD focuses evaluation on salient, detail-rich regions and avoids the pitfalls of downsampling-based scoring, offering a more faithful measure of legibility and fine-grain preservation.

In summary, our contributions are fourfold: (1) **Position-Refined Positional Encoding (PRPE).** We remap positional encodings using garment–body correspondences to inject alignment-aware spatial cues into DiT attention, markedly improving spatial accuracy and high-frequency texture fidelity *without* introducing extra modules or bespoke loss terms on diffusion features. (2) **Positional Relation Learning (PRL).** A lightweight module that learns a stable, dense remapping field directly from in-context pairs, eliminating the need for precomputed matches or manual labels. (3) **High-resolution evaluation protocol.** We expose the shortcomings of common metrics and introduce **loc-CMMD**; together with stronger perceptual backbones (**CMMD**, **DINOv2-FD**, **Dream-Sim**), this protocol more faithfully assesses global consistency and the legibility of textual logos and micro-textures. (4) **Extensive empirical validation.** Extensive experiments show that PR-DIT outperforms prior state-of-the-art on both conventional and newly proposed metrics, yielding clear gains in visual quality and detail preservation.

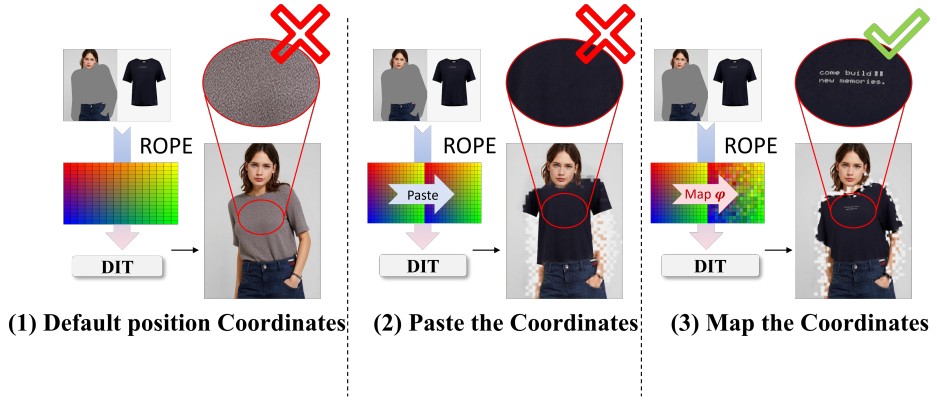

**(1) Default position Coordinates**    **(2) Paste the Coordinates**    **(3) Map the Coordinates**

Figure 2: In the ICL setting, zero-shot virtual try-on with an inpainting model: (1) Default PE assigns a single coordinate frame to both the masked image and the garment, often causing the model to ignore the reference and hallucinate an incorrect outfit. (2) Coordinate copy duplicates the masked-image coordinates onto the garment; tokens at identical positions attend to each other, yielding a direct "paste-in" effect. (3) Position mapping $\phi$ sets equal coordinates only for true correspondences between the masked image and the garment, enabling zero-shot synthesis that preserves the garment's original fine details while remaining spatially plausible.

## 2 RELATED WORK

### 2.1 APPEARANCE-FLOW–BASED TWO-STAGE SYNTHESIS

Early virtual try-on systems lacked powerful generative backbones and therefore adopted *two-stage, appearance-flow* pipelines: a warping network first predicted a location mapping (appearance flow)

conditioned on preprocessed human cues (pose, parsing), transforming the reference garment to match the target body; a second-stage compositor (typically a GAN) then rendered the warped garment onto the person. Representative works included VITON Han et al. (2018), which pioneered the "geometry alignment + synthesis" paradigm with a garment-agnostic person representation and a coarse-to-fine pipeline; ACGPN Yang et al. (2020), which predicted semantic layouts before warping and content refinement, and switched adaptively between generation and preservation to boost realism; and PF-AFN Ge et al. (2021), which introduced progressive flow alignment to stabilize multi-scale deformation. More recently, GP-VTON Xie et al. (2023) decomposed warping into torso and left/right sleeves and fused a global parser (LFGP) to better retain textures and text under complex poses; D4-VTON Yang et al. (2024) performed grouped/block-wise warping in feature space and used a differentiable grouping strategy to ease semantic mismatch and reduce reliance on static parsing. While such appearance-flow methods explicitly provided the desired position mapping, they came with substantial engineering overhead and error accumulation across stages. Compared to these deformation-centric pipelines that enforced text/texture fidelity via explicit warps, our approach preserves correspondence without explicit garment warping: we refine positional encodings to *modulate* attention, thereby improving high-resolution detail fidelity within a single diffusion-based generation process.

### 2.2 SINGLE-STAGE DIFFUSION–BASED VIRTUAL TRY-ON

With the advent of strong diffusion backbones, the field shifted to *single-stage, diffusion-based* try-on that treated the garment as an image condition injected into the generator. TryOnDiffusion Zhu et al. (2023) systematized diffusion for try-on with a dual-UNet pipeline and delivered high-fidelity synthesis. IDM-VTON Choi et al. (2024) introduced a reference-net branch to inject garment-detail features and improved robustness under real, cluttered backgrounds. CatVTON Chong et al. (2024) advocated the "concatenate-and-generate" ICL paradigm, spatially stacked person and garment inputs, and fine-tuned self-attention to reach SOTA. FitDiT Jiang et al. (2024) brought DiT architectures to try-on and pretrained a garment texture extractor to strengthen detail injection. Beyond plain conditioning, newer approaches aimed to *reference the right locations*: SPM-Diff Wan et al. (2025) encoded fine appearance as structured semantic point sets and matched garment-to-person correspondences via local manifold warps; Leffa Zhou et al. (2025) learned an attention "flow" under direct supervision of query-to-reference focus, thereby mitigating detail distortion. Our method is also correspondence-centric, but instead of introducing explicit warping modules or heavy supervision on attention maps, we exploit the inductive role of positional encoding in DiT: we *refine* RoPE Su et al. (2024) coordinates to steer attention toward geometry-consistent matches, requiring no extra complex modules or losses while achieving high-fidelity synthesis at high resolution.

## 3 METHODOLOGY

### 3.1 PRELIMINARY

**In-Context Learning.** Our core innovation lies in optimizing positional encoding (PE). Given that in-context learning (ICL) inherently aligns features across concatenated condition images, it optimally leverages position-guided information. Consequently, we adopt an ICL-based approach as our baseline to exploit this positional guidance effectively.

Virtual try-on is defined as follows: given a person image $\mathbf{I}_p \in \mathbb{R}^{H \times W \times 3}$, a target garment-region mask $\mathbf{M}_g \in \mathbb{R}^{H \times W \times 1}$, and a reference garment image $\mathbf{I}_g \in \mathbb{R}^{H \times W \times 3}$, the objective is to synthesize a try-on image $\mathbf{I}_t \in \mathbb{R}^{H \times W \times 3}$. In in-context learning (ICL), we first apply the person mask $\mathbf{M}_p$ to $\mathbf{I}_p$ to generate a garment-agnostic person image $\mathbf{I}_m$. Subsequently, $\mathbf{I}_m$ and $\mathbf{I}_g$ are concatenated along the spatial dimension and, under a fixed text prompt, fed into a diffusion-based inpainting model, which outputs $\mathbf{D}' = [\mathbf{I}_t, \mathbf{I}'_g]$, where $\mathbf{I}_t$ is the final result. Within the Diffusion Transformer (DiT) backbone, $\mathbf{I}_g$ and $\mathbf{I}_m$ are encoded into feature maps $\mathbf{F}_g$ and $\mathbf{F}_m$, respectively, which are then processed through $L$ Transformer blocks.

**Positional Encodings in DiT.** Unlike convolutional diffusion models, such as Stable Diffusion, which implicitly maintain spatial relationships through local connectivity, Diffusion Transformers (DiTs) are permutation-invariant and thus necessitate explicit positional encodings. The FLUX fam-

ily employs Rotary Position Embeddings (RoPE), which incorporate relative positional information into the attention mechanism by applying position-dependent rotations to the query and key vectors.

As illustrated in Figure 3, the queries and keys are derived from learned linear projections of the token features (e.g., from $[\mathbf{F}g, \mathbf{F}m]$). RoPE then applies a rotation operator $\mathcal{R}\Theta(\cdot)$ whose phase depends on the token position, ensuring that attention scores are modulated by *relative* offsets rather than absolute coordinates and thereby imparting a translation-friendly inductive bias. Concretely, $\Theta = \theta_i i = 1^{d/2}$, where $\theta_i = 10000^{-2(i-1)/d}$. The rotation matrix for integer position $p$ is

$$\mathcal{R}_\Theta(p) = \mathrm{diag}\left( \begin{bmatrix} \cos(p\theta_1) & -\sin(p\theta_1) \\ \sin(p\theta_1) & \cos(p\theta_1) \end{bmatrix}, \ldots, \begin{bmatrix} \cos(p\theta_{d/2}) & -\sin(p\theta_{d/2}) \\ \sin(p\theta_{d/2}) & \cos(p\theta_{d/2}) \end{bmatrix} \right). \tag{1}$$

In FLUX.1-dev, each token position is represented as a 3-D vector $[p, x, y]$. Multi-axis RoPE applies independent rotations in disjoint channel subspaces corresponding to these axes. Text tokens typically use $[0, 0, 0]$ (the text encoder provides their semantics), whereas image tokens use $t = 0$ with $(x, y)$ taken from the token grid (either the patch grid or the latent grid). Since $p$ is fixed to 0 in our setting, all subsequent analysis focuses on the spatial coordinates $[x, y]$. As illustrated in the RoPE module of Figure 3, we denote this 2-D spatial portion of the positional encoding by $\mathcal{C} = (x, y)$, with $\mathcal{C} \in \Omega$ and $\Omega = \{0, \ldots, H-1\} \times \{0, \ldots, W-1\}$, and $\mathcal{C}$ can be viewed as a coordinate field over the $H \times W$ token grid.

## 3.2 POSITION-REFINED POSITIONAL ENCODING

**Problem Statement and Learning Objective.** As explained in the introduction and illustrated in Figure 2, assigning default coordinates either by vertical concatenation, $\mathcal{C}_g^0(i, j) = (i, j)$ and $\mathcal{C}_p^0(i, j) = (i+H, j)$, or by simple pasting, $\mathcal{C}_g^0(i, j) = (i, j)$ and $\mathcal{C}_p^0(i, j) = (i, j)$, fails to encode the true local semantic correspondences between $\mathbf{I}_g$ and $\mathbf{I}_m$. In contrast, correcting positional encodings with a proper garment→person mapping markedly improves alignment. Since the synthesis target is the person branch $\mathbf{I}_m$, we keep its coordinate field as the default grid $\mathcal{C}_p(i, j) = (i, j)$, while the garment coordinates are obtained via the position mapping. Our goal is therefore to estimate the following position mapping:

$$\phi: \ \Omega \to \mathbb{R}, \quad (u, v) \mapsto (i, j), \qquad \Omega = \{0, \ldots, H-1\} \times \{0, \ldots, W-1\}, \tag{2}$$

which assigns to each garment coordinate $(u, v)$ its semantically corresponding person coordinate $(i, j)$. We use $\phi$ to modulate RoPE so that attention peaks at geometrically correct correspondences. Once $\phi$ is estimated, we construct the mapped garment coordinate field $\mathcal{C}_g(u, v) = \phi(u, v)$, RoPE then rotates queries/keys using these coordinates:

$$\tilde{\mathbf{q}}_{i,j}^{(p)} = \mathcal{R}_\Theta\big(\mathcal{C}_p(i, j)\big) \mathbf{q}_{i,j}^{(p)}, \qquad \tilde{\mathbf{k}}_{u,v}^{(g)} = \mathcal{R}_\Theta\big(\mathcal{C}_g(u, v)\big) \mathbf{k}_{u,v}^{(g)}. \tag{3}$$

By the relative-phase property of RoPE, the cross-attention score becomes

$$\alpha_{(i,j)\to(u,v)} \ \propto \ \big\langle \mathbf{q}_{i,j}^{(p)}, \ \mathcal{R}_\Theta\big(\mathcal{C}_g(u, v) - \mathcal{C}_p(i, j)\big) \mathbf{k}_{u,v}^{(g)} \big\rangle. \tag{4}$$

When $(u, v)$ matches $(i, j)$, $\mathcal{C}_g(u, v) \approx \mathcal{C}_p(i, j)$; the phase tends to zero, $\mathcal{R}_\Theta$ becomes identity, and attention is amplified. Mismatches yield nonzero phase and suppress the score. Thus semantic alignment is enforced directly via positional encoding, without extra losses, schedules, or architectural changes. Next, we present two simple estimators for the mapping $\phi$.

**Flow Guided Position Mapping.** As shown in Figure3(1). Appearance flow naturally encodes the spatial mapping needed by virtual try-on. A pretrained warping module $\mathcal{W}$ predicts a forward flow $\mathbf{F}: \Omega \to \mathbb{R}^2$, $\mathbf{F}(i, j) = (u, v)$, meaning that a person location $(i, j)$ corresponds to the continuous garment coordinate $(u, v)$. Denote $\mathbf{T}(i, j) = \mathbf{F}(i, j)$. We approximate the inverse map $\tilde{\mathbf{T}}^{-1}: \Omega \to \mathbb{R}^2$ via scattered interpolation on $\mathbf{F}$: for each garment grid point $(u, v) \in \Omega$, find its enclosing forward-flow neighbors on $\Omega$ and compute the corresponding person coordinate $(\hat{i}, \hat{j}) \in \mathbb{R}^2$. Then define the garment coordinate field: $\phi_{\text{flow}}(u, v) = \tilde{\mathbf{T}}^{-1}(u, v) = (\hat{i}, \hat{j}) \in \mathbb{R}^2$.

**Mask Guided Position Mapping.** As shown in Figure3(2), we derive a coarse cloth-to-person correspondence directly from the binary masks of the garment $\mathcal{M}_g$ and the person $\mathcal{M}_p$. We first

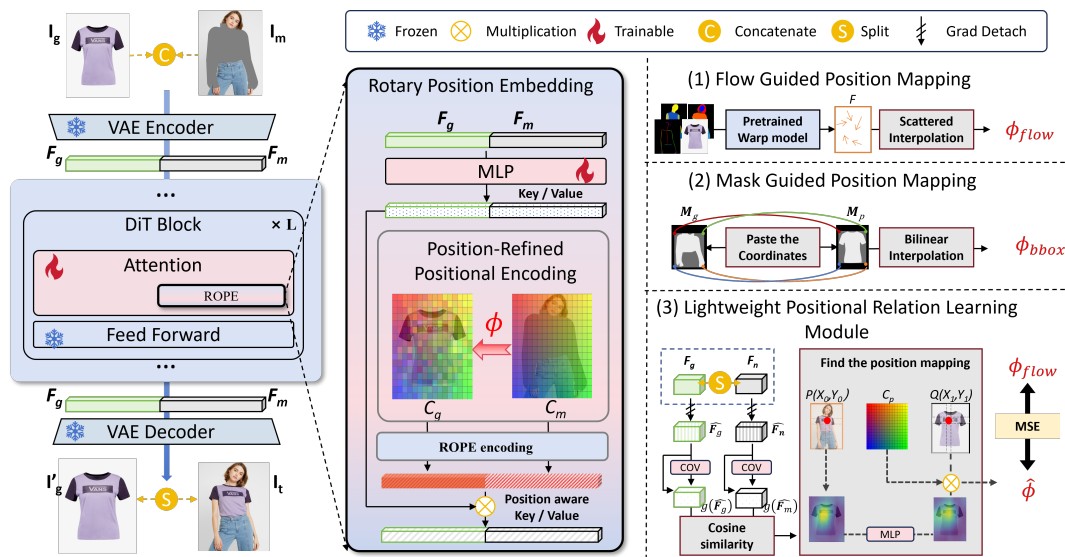

Figure 3: Overview of PR-VTON. Left: Within the DiT backbone we modify only the RoPE-based positional mapping, underscoring the method's plug-and-play usability. Before attention, keys/queries are RoPE-encoded from per-token coordinates; we keep the masked-person coordinates $C_m$ at the default grid and set the garment coordinates $C_g$ using the garment-to-person mapping $\phi$. Right: Estimating $\phi$. (1) *Flow-guided*: the precomputed appearance flow encodes correspondences; we fit $\phi$ via scattered/bilinear interpolation. (2) *Mask-guided*: take the axis-aligned bounding-box corners of $M_g$ and $M_p$ as correspondences and interpolate to obtain a coarse field. (3) *Lightweight PRL*: a positional-relation learning module that infers $\phi$ from the intrinsic affinity of semantically matched tokens.

compute the axis-aligned bounding boxes for both masks and enforce a corner-to-corner alignment (top-left, top-right, bottom-left, bottom-right). For any garment pixel inside its box, we normalize its coordinates to the unit square and obtain its target location in the person box via *bilinear interpolation* of the four aligned corners. The resulting mapping $\phi_{\mathrm{bbox}}$ is a dense, backbone-agnostic, training-free coordinate field. Although coarse, it reliably steers attention toward globally correct matches and serves as a lightweight prior for RoPE modulation.

## 3.3 LIGHTWEIGHT POSITIONAL RELATION LEARNING (PRL) MODULE VIA DIFFUSION FEATURE AUTOCORRELATION

While flow-PE achieves fine local alignment, it relies on an external optical-flow module; conversely, bbox-PE is straightforward to implement but yields only coarse correspondences. We thus pursue a method that enables fine-grained semantic alignment without any external dependencies. Drawing inspiration from DIFT Tang et al. (2023), we recognize that features within diffusion models inherently encode rich semantics; however, DIFT emphasizes sparse keypoint matches, whereas we require dense positional guidance. To this end, we introduce a self-supervised strategy that estimates the mapping $\phi$ solely from the *autocorrelation* between diffusion features, yielding $\phi$ without auxiliary modules.

Concretely (cf. Figure3(3)), at the $K$-th DiT block we take the garment tokens $\boldsymbol{F}_g$ and the masked-person tokens $\boldsymbol{F}_m$. To use semantics independently and prevent the mapping branch from affecting generation quality, we *stop gradients* when computing $\phi$: $\hat{\boldsymbol{F}}_g = \mathrm{sg}(\boldsymbol{F}_g)$, $\hat{\boldsymbol{F}}_m = \mathrm{sg}(\boldsymbol{F}_m)$, where $\mathrm{sg}(\cdot)$ denotes stop-gradient (i.e., gradient detachment). To enlarge the receptive field and obtain discriminative local semantics, we define a lightweight covariance augmentation $g(\boldsymbol{x}) = \mathrm{cov}(\boldsymbol{x}) + \boldsymbol{x}$ and apply it to both branches, yielding $g(\hat{\boldsymbol{F}}_g)$ and $g(\hat{\boldsymbol{F}}_m)$. We then form a cosine-similarity matrix $\mathbf{H} = \cos\big(g(\hat{\mathbf{F}}_g), g(\hat{\mathbf{F}}_m)\big) \in \mathbb{R}^{(HW)\times(HW)}$. Figure3(3) details how to derive the corresponding positional mapping from $H$. Taking a point $P(x_0, y_0)$ on the masked image as an example, we explain how to find its mapped point $Q(x_1, y_1)$: first, extract the heatmap $H(x_0, y_0)$ of size $h \times w$, which rep-

resents the similarity of each point on the garment to $P$; next, apply a row-wise MLP for refinement, followed by softmax over the person position dimension: $\tilde{H}_{u,(i,j)} = \text{softmax}_{(i,j)}\big(\phi_{\text{MLP}}(H_{u,(:)})\big)$, producing the optimized positional similarities.

Let $\boldsymbol{C}_p \in \mathbb{R}^{(HW) \times 2}$ stack all person grid coordinates $(i, j)$. The predicted garment $\rightarrow$person mapping is the similarity-weighted centroid (our $\phi$):

$$\boldsymbol{C}_{\text{pred}}(u) = \sum_{(i,j)} \tilde{\boldsymbol{H}}_{u,(i,j)}\,(i, j) = \tilde{\boldsymbol{H}}_{u,:}\,\boldsymbol{C}_p, \qquad \phi(u) \triangleq \boldsymbol{C}_{\text{pred}}(u). \tag{5}$$

We supervise $\phi$ with dense pseudo ground truth $\phi_{\text{flow}}$ provided by an appearance-flow estimator using a simple regression loss:

$$\mathcal{L}_{\text{flow}} = \frac{1}{|\Omega|} \sum_{u \in \Omega} \big\|\phi(u) - \phi_{\text{flow}}(u)\big\|_2^2, \qquad \Omega = \{0, \ldots, H-1\} \times \{0, \ldots, W-1\}. \tag{6}$$

Finally, the predicted coordinates directly *modulate RoPE* by rotating garment-side tokens at their mapped person locations:

$$\text{PE}_{\text{pred}}(u) = \mathcal{R}_\Theta\big(\boldsymbol{C}_{\text{pred}}(u)\big), \tag{7}$$

after which queries/keys are rotated as in standard RoPE. Throughout the model, we use the bbox-guided mapping $\phi_{\text{bbox}}$ for all layers before $K$; at layer $K$ we switch to online estimation of the mapping and reuse it for the remaining $L-K$ layers (i.e., for all $\ell \geq K$). In this way, the network first follows a stable coarse prior $\phi_{\text{bbox}}$ and then self-corrects to a fine-grained mapping $\phi$. In this way, semantic alignment is realized through diffusion-internal feature autocorrelation and encoded at the positional-embedding level—fine-grained and external-module-free.

## 3.4 TRAINING STRATEGY

Training PRL from scratch can be unstable when the early estimate $\widehat{\phi}$ is inaccurate. We therefore adopt a progressive schedule tightly coupled to the mapping objective. During the first $T$ training steps, we modulate the DiT with the flow-warped positional encoding ("Flow-warped PE", Sec. 3.2), i.e., set $\phi \approx \phi_{\text{flow}}$, which provides reliable spatial guidance while the generator learns the base task. After step $T$, we switch the positional encoding in layers $\ell \geq k$ to the network's own prediction $\widehat{\phi}$ (via $\text{PE}_{\text{pred}}$), aligning training with inference and enabling end-to-end refinement.

Let $\mathcal{L}_{\text{FM}}$ denote the flow-matching denoising loss. The total objective is

$$\mathcal{L} = \mathcal{L}_{\text{FM}} + \lambda\,\mathcal{L}_{\text{PE}}, \tag{8}$$

where $\mathcal{L}_{\text{PE}}$ supervises the predicted mapping $\widehat{\phi}$ (e.g., against $\phi_{\text{flow}}$ or a coarse prior).

## 4 EXPERIMENTS

### 4.1 IMPLEMENTATION DETAILS AND DATASETS

We adopt **FLUX.1-Fill-dev** as the backbone, and use GP-VTON Xie et al. (2023) as the appearance-flow predictor in 3.2. Following prior work, we evaluate on the standard VITON-HD Choi et al. (2021) (13,679 images—11,647 train / 2,023 test; upper body) and DressCodeMorelli et al. (2022) (53,792 images—48,392 train / 5,400 test; tops/bottoms/dresses) benchmarks. Unless otherwise stated, we use a batch size of 16 and train for 100,000 steps on both datasets. We fine-tune only the *attention layers* of FLUX and our *lightweight Positional Relation Learning (PRL) module*, keeping all other parameters frozen. All experiments are conducted on eight NVIDIA H800 GPUs with a learning rate of $2 \times 10^{-5}$.

Within the TransformerBlock, we select layers $k \in \{40, 41, 42\}$ for positional-encoding correction and take the average of the three predicted PEs as the final estimate. The loss trade-off coefficient is set to $\lambda = 10^{-2}$, and the schedule switch step is $T = 5\text{K}$.

### 4.2 LIMITATIONS OF EXISTING EVALUATION METRICS

Table 1: Tailored metrics on VITON-HD. CMMD/DINOv2-FD/DreamSim reflects global agreement; OCR-FD targets high-frequency detail in both paired and unpaired setups.

| Methods | Paired | | | | | Unpaired | |
|---|---|---|---|---|---|---|---|
| | CMMD↓ | DINOv2↓ | DreamSim↓ | Loc-CMMD$_A$↓ | Loc-CMMD$_O$↓ | CMMD↓ | DINOv2↓ |
| Leffa | 0.042 | 22.388 | 0.0339 | 0.058 | 0.091 | 0.054 | 54.853 |
| IDM-VTON | 0.131 | 30.096 | 0.0433 | 0.151 | 0.275 | 0.138 | 58.311 |
| Any2AnyTryon | 0.154 | 94.125 | 0.0647 | 0.203 | 0.236 | 0.187 | 130.839 |
| FitDiT | 0.087 | 55.209 | 0.0587 | 0.190 | 0.175 | 0.092 | 79.308 |
| CatVTON-Flux | 0.029 | 26.055 | 0.0372 | 0.059 | 0.073 | 0.038 | 58.427 |
| Warp-and-Paste | 0.158 | 34.260 | 0.0361 | 0.303 | 0.481 | - | - |
| **Ours** | **0.013** | **20.305** | **0.0293** | **0.045** | **0.051** | **0.022** | **20.367** |

A critical step toward progress in virtual try-on is establishing evaluation metrics that reflect human perception. However, widely used measures—FID Parmar et al. (2022), KID Bińkowski et al. (2018), SSIM Wang et al. (2004), and LPIPS Zhang et al. (2018)—fail to capture two essentials for high-resolution synthesis: (*i*) **global perceptual agreement** (holistic fit and visual coherence) and (*ii*) **fine-grained detail fidelity**.

To reveal the mismatch on global perception, we design a simple **Warp-and-Paste** baseline that involves no generative modeling.

Table 2: Conventional-metric comparison on VITON-HD. The best and 2nd-best are highlighted in bold and underlined formats.

| Methods | Paired | | | | Unpaired | |
|---|---|---|---|---|---|---|
| | SSIM↑ | FID↓ | KID↓ | LPIPS↓ | FID↓ | KID↓ |
| GP-VTON Xie et al. (2023) | 0.894 | 9.20 | 3.94 | 0.080 | 11.84 | 4.31 |
| LaDI-VTON Morelli et al. (2023) | 0.876 | 6.66 | 1.08 | 0.091 | 9.41 | 1.60 |
| IDM-VTON Choi et al. (2024) | 0.870 | 6.29 | 0.73 | 0.102 | 9.84 | 1.12 |
| OOTDiffusion Xu et al. (2025) | 0.878 | 8.81 | 0.82 | 0.071 | 12.41 | 4.69 |
| CatVTON Chong et al. (2024) | 0.870 | 5.43 | 0.41 | 0.057 | 9.02 | 1.09 |
| LeFFA Zhou et al. (2024) | 0.899 | 4.54 | **0.05** | **0.048** | 8.52 | **0.32** |
| FitDiT Jiang et al. (2024) | 0.899 | 4.73 | 0.19 | 0.066 | **8.20** | 0.34 |
| Any2AnyTryon Guo et al. (2025) | 0.839 | 6.93 | 0.74 | 0.087 | 8.97 | 0.98 |
| SPM-Diff Wan et al. (2025) | 0.917 | 6.87 | 0.52 | 0.055 | - | - |
| MV-VTON Wang et al. (2025) | 0.897 | 5.43 | 0.49 | 0.069 | 17.90 | 8.86 |
| ITA-MDT Hong et al. (2025) | 0.885 | 5.46 | - | 0.083 | 8.68 | - |
| warpe and paste | **0.923** | 4.84 | 0.322 | 0.0478 | - | - |
| baseline | 0.875 | 5.50 | 0.54 | 0.069 | - | - |
| Ours | 0.884 | **4.35** | 0.10 | 0.051 | 8.49 | 0.55 |

Given a garment image $I_c$ and a person image $I_m$, we warp $I_c$ using the appearance flow $F$ (see Sec. 3) and composite it onto $I_m$:

$$I_{\text{pred}} = \underbrace{\text{grid\_sample}(I_c; F)}_{\text{warped garment}} \odot M_p \ \oplus \ I_m \odot (1 - M_p), \qquad (9)$$

where $M_p$ is the warped garment foreground mask; $\odot$ and $\oplus$ denote element-wise multiplication and addition, respectively. The outputs clearly exhibit seam artifacts, lack shading, and look "pasted-on." *Paradoxically*, Table 2 shows competitive—or even superior—scores under SSIM, KID, and LPIPS, exposing a severe disconnect between these metrics and perceptual plausibility.

For detail consistency, most metrics resize images before feature extraction (e.g., FID uses an Inception-V3 Szegedy et al. (2016) encoder at $299{\times}299$), which irreversibly discards high-frequency content critical to try-on (brand logos, fine patterns, small text). This downsampling renders the metrics insensitive to local degradations, failing to distinguish high-quality text from distorted replicas in high-resolution outputs.

Table 3: Tailored metrics on DressCode. CMMD/DINOv2-FD/DreamSim reflect global agreement; OCR-FD targets high-frequency detail in both paired and unpaired setups.

| Methods | Paired | | | Unpaired | |
|---|---|---|---|---|---|
| | CMMD↓ | DINOv2↓ | DreamSim↓ | CMMD↓ | DINOv2↓ |
| IDM-VTON | 0.087 | 36.291 | 0.0395 | 0.097 | 55.804 |
| Any2AnyTryon | 0.053 | 57.280 | 0.0805 | 0.073 | 95.425 |
| FitDiT | 0.050 | 46.432 | 0.0485 | 0.053 | 61.253 |
| **Ours** | **0.021** | **18.260** | **0.0321** | **0.037** | **53.068** |

### 4.3 Metrics Tailored for Virtual Try-On

Motivated by the above shortcomings, we adopt metrics that better reflect *global* coherence and *local* detail fidelity. For global consistency, inspired by Dream-Sim, we compute image-level perceptual alignment using strong vision encoders—CMMD (CLIP-based) Jayasumana et al. (2024), FD-DINOv2 Stein et al. (2023); Ahn (2024), and DreamSim Fu et al. (2023); Sundaram et al.

Table 4: Conventional-metric comparison on DressCode.

| Methods | Paired | | | | Unpaired | |
| --- | --- | --- | --- | --- | --- | --- |
| | SSIM↑ | FID↓ | KID↓ | LPIPS↓ | FID↓ | KID↓ |
| GP-VTON Xie et al. (2023) | 0.771 | 9.93 | 4.61 | 0.180 | 12.79 | 6.63 |
| LaDI-VTON Morelli et al. (2023) | 0.906 | 4.14 | 1.21 | 0.064 | 6.49 | 2.20 |
| IDM-VTON Choi et al. (2024) | 0.920 | 8.64 | 2.92 | 0.062 | 9.55 | 4.32 |
| OOTDiffusion Xu et al. (2025) | **0.927** | 4.20 | 0.37 | 0.045 | 12.57 | 6.63 |
| CatVTON Chong et al. (2024) | 0.892 | 3.99 | 0.82 | 0.046 | 6.14 | 1.40 |
| FitDiT Jiang et al. (2024) | 0.926 | 2.64 | 0.50 | 0.043 | **4.73** | **0.90** |
| Ours | 0.909 | **2.12** | **0.06** | **0.035** | 4.93 | 1.08 |

(2024)—which capture semantics, layout, and overall visual coherence more faithfully than the Inception features used by FID. For local details, to avoid the downsampling inherent in many metrics, we evaluate directly on high-resolution, detail-rich regions via our **loc-CMMD**; implementation specifics are provided in the appendix.

## 4.4 QUANTITATIVE RESULTS

Tables 1,3 show state-of-the-art results under SSIM/FID/KID/LPIPS in both paired and unpaired settings; notably, simply correcting the positional encoding already yields large gains over our Baseline. 2,4summarize perception-aligned results on VITON-HD: Loc-CMMD$_A$ evaluates the full set, Loc-CMMD$_O$ targets text garments. Using public checkpoints for representative SD/DiT baselines, our method ranks first with larger margins than conventional metrics; the heuristic warp-and-paste scores much worse on CMMD/DINOv2, and the Loc-CMMD$_O$ gap widens—especially on Loc-CMMD$_{AA}$. See Appendix for details.

## 4.5 ABLATION STUDIES.

We ablate the layer-scheduling hyperparameter $k$ at 768 resolution for efficiency. As reported in Table 5, $k = 40$ delivers the best overall scores. When $k$ is too small, masked and reference tokens have insufficient affinity, leading to unreliable semantic correspondences. When $k$ is too large, most blocks remain driven by the coarse $\phi_{\text{bbox}}$ prior, limiting detail

Table 5: Ablation Studies.

| | SSIM ↑ | FID↓ | KID ↓ | LPIPS↓ |
| --- | --- | --- | --- | --- |
| k=10 | 0.8700 | 5.34 | 0.332 | 0.0603 |
| k=30 | 0.869 | 5.32 | 0.348 | 0.0609 |
| k=40 | 0.870 | 5.28 | 0.259 | 0.0597 |
| k=50 | 0.868 | 5.39 | 0.259 | 0.0617 |

fidelity. A mid-range $k$ thus strikes the right balance between coarse guidance and learned refinement.

## 5 CONCLUSION

We presented **PR-VTON**, a simple yet effective framework that improves high-frequency detail preservation in virtual try-on by acting directly on positional encodings. Our **PRPE** remaps RoPE coordinates using garment–body correspondences to inject alignment-aware cues into DiT attention, while the lightweight **PRL** module learns a dense remapping field from in-context pairs, avoiding extra modules or bespoke losses. A zero-shot ICL analysis motivated this design and showed that precise positional correspondences alone can steer attention to aggregate reference textures along semantically correct paths. Beyond methodology, we revisited evaluation and introduced **loc-CMMD** alongside stronger perceptual backbones (CMMD, DINOv2-FD, DreamSim) to better capture global coherence and text/texture legibility at high resolution. Extensive experiments on `VITON-HD` and `DressCode` demonstrate that our DiT instantiation (**PR-DiT**) surpasses prior work on both conventional and perception-aligned metrics, yielding clear gains in visual quality and fine-detail fidelity.

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

## A   APPENDIX

### A.1   LOC-CMMD: IMPLEMENTATION DETAILS.

Concretely, we first run PP-OCRv5 Cui et al. (2025) on the ground-truth image $I_{gt}$ to detect text regions and select the highest-confidence bounding box with score $> 0.8$; we require the box center to lie within the person mask $M_p$, and if no candidate satisfies this, we fall back to the geometric centroid of $M_p$ (which typically lies near the garment's texture center). We then crop a $225 \times 225$ patch around this center on $I_{gt}$ and take the co-located patch on $I_{pred}$, and compute the DINOv2-based similarity (DINOv2-FD) on these crops. This cropping protocol sidesteps global resizing, preserves fine textures (logos, dense patterns, small text), and yields a more discriminative measure of detail fidelity.

### A.2   EXPERIMENTAL ANALYSIS.

Table2 reports results on perception-aligned metrics that better reflect holistic agreement and detail fidelity on VITON-HD. Here, Loc-CMMD$_A$ denotes evaluation over the entire test set, while Loc-CMMD$_O$ restricts evaluation to garments with detected text regions. For fair comparison, we select representative methods from both SD-based and DiT-based families, using their public checkpoints for inference: **LeFFA** and **IDM** (SD-base), and **Any2Any** and **FitDiT** (DiT-base). Our method attains the best performance on all these metrics, with margins that are *larger* than those observed under conventional metrics—evidence that the new measures are more sensitive to perceptual quality. In particular, the heuristic *warp-and-paste* baseline scores markedly worse on **CMMD** and **DI-NOv2** than methods with superior visual plausibility (e.g., LeFFA, Baseline, IDM), a contrast that conventional metrics fail to expose. On our proposed Loc-CMMD family, the gap further widens in favor of our approach; the difference on Loc-CMMD$_O$ exceeds that Loc-CMMD$_A$, indicating

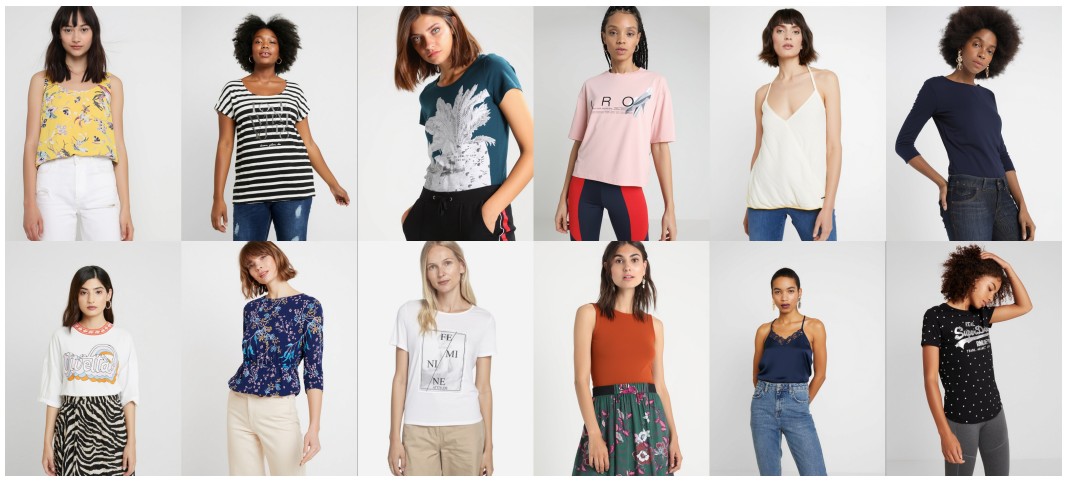

Figure 4: Our generation effect.

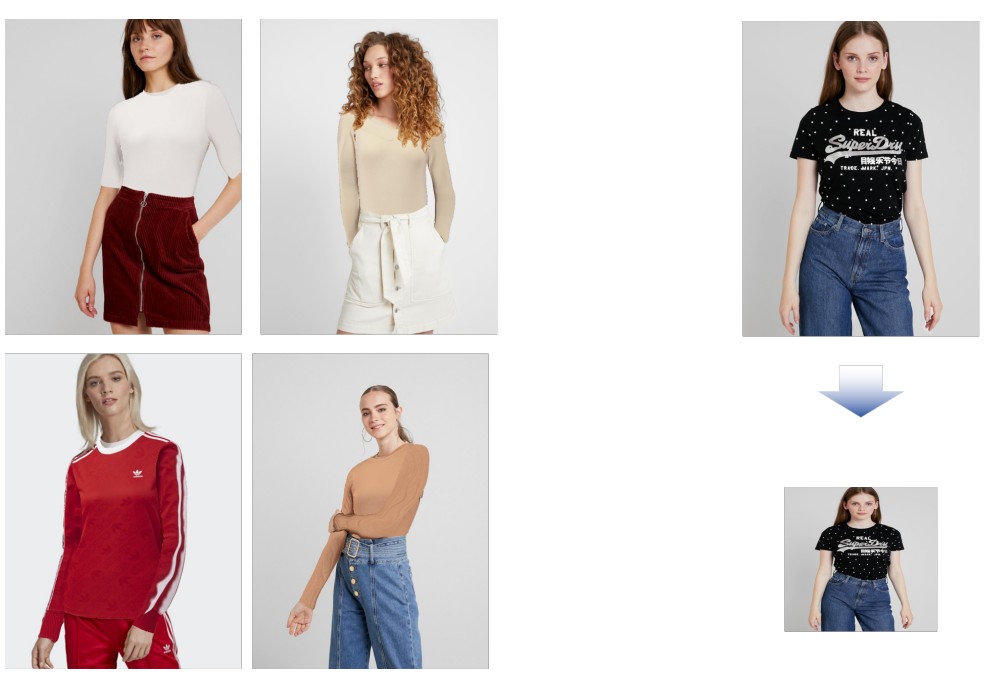

(1) warp and paste

(2) resize to 299*299

Figure 5: warpe and paste, resize

that modulating positional encoding confers a pronounced advantage in synthesizing fine, text-like details.

