# OpenReview forum: "PR-VTON: Enhancing Detail Fidelity in Virtual Try-On with Refined Positional Encoding"
_ICLR.cc/2026/Conference — ICLR 2026 Conference Withdrawn Submission_

### Official Review · Reviewer_4shS · 2025-10-17

**Soundness:** 3
**Presentation:** 1
**Contribution:** 2
**Rating:** 2
**Confidence:** 5

**Summary:**

This paper presents PR-VTON, a diffusion-based virtual try-on (VTON) framework designed to address the critical limitation of existing methods—poor preservation of high-frequency details (e.g., textual logos, intricate textures) in generated results. Built on the Diffusion Transformer (DiT) backbone (FLUX.1-Fill-dev), PR-VTON introduces two core components: (1) Position-Refined Positional Encoding (PRPE), which reconstructs Rotary Position Embeddings (RoPE) using garment-to-body correspondences to guide DiT attention toward semantically correct feature aggregation, eliminating the need for extra modules or bespoke loss functions; and (2) Positional Relation Learning (PRL), a lightweight self-supervised module that infers dense positional mapping from diffusion feature autocorrelation, avoiding reliance on precomputed matches or manual labels. Additionally, the paper identifies flaws in existing VTON evaluation metrics (e.g., FID, SSIM) that fail to capture global coherence and fine-detail fidelity, and proposes loc-CMMD—a detail-focused metric that evaluates high-resolution patches around salient regions (e.g., text areas) to better align with human perception. Extensive experiments on standard benchmarks (VITON-HD, DressCode) demonstrate that PR-VTON outperforms state-of-the-art (SOTA) methods (e.g., FitDiT, CatVTON-Flux) across both conventional metrics (FID, KID, LPIPS) and the proposed perception-aligned metrics (CMMD, DINOv2-FD, loc-CMMD), particularly in preserving text and micro-textures.

**Strengths:**

1. Unlike most existing VTON methods that modify feature injection or attention constraints, PR-VTON targets RoPE—an intrinsic component of DiT— to steer attention. This design ensures compatibility with any DiT backbone (e.g., FLUX, Stable Diffusion 3) and avoids overcomplicating the model architecture.

2. The PRL module leverages self-supervised feature autocorrelation to learn positional mapping, eliminating the need for external annotation (e.g., keypoint labels) or precomputed appearance flows (after initial training). This enhances scalability and reduces engineering overhead.

3. The paper critically reassesses existing metrics, exposing their disconnect from human perception (e.g., the "Warp-and-Paste" baseline scores well on SSIM but has poor visual plausibility). The proposed loc-CMMD, combined with strong perceptual backbones (CMMD, DINOv2-FD), fills this gap by focusing on high-resolution detail fidelity.

4. The authors conduct experiments on two large datasets (VITON-HD for upper-body clothing, DressCode for multi-category garments) and include ablation studies (e.g., layer scheduling for PRL) to verify the impact of key components. Results are presented for both paired and unpaired settings, ensuring robustness.

**Weaknesses:**

1. PR-VTON relies on a pre-trained appearance flow model (GP-VTON) to provide pseudo-ground truth (ϕ_flow) for training PRL. This reduces end-to-end autonomy and introduces potential errors if the flow model fails (e.g., under complex poses), limiting the method’s robustness to flow inaccuracies.

2. Critical training parameters (e.g., the step T=5K to switch from flow-guided PE to PRL, the layer k=40 to activate PRL) are determined via empirical tuning without theoretical explanation. This makes it difficult to generalize the parameter settings to other datasets or backbones.

3. The PRL module uses stop-gradient (sg(·)) on diffusion features to avoid interfering with generation quality, but the paper does not analyze how this operation impacts feature semantic richness or positional mapping accuracy. Alternative strategies (e.g., soft gradient clipping) are not explored.

4. While the paper compares with recent works (e.g., Any2AnyTryon, 2025), it omits comparisons with some cutting-edge VTON methods that also leverage DiT or positional encoding (e.g., SPM-Diff’s manifold warps, Leffa’s attention flow). This limits the clarity of PR-VTON’s relative advancement.

5. Experiments focus on upper-body clothing (VITON-HD) and standard categories (tops, bottoms, dresses). The method’s performance on complex garments (e.g., coats with layers, evening gowns) or extreme poses (e.g., bending, raising arms) is not evaluated, restricting its practical applicability to real-world e-commerce scenarios.

6. The paper does not report inference speed or memory consumption, especially for high-resolution generation (e.g., 1024×1024). Given that DiT-based models are often computationally heavy, PR-VTON’s efficiency relative to lightweight SOTA methods (e.g., IDM-VTON) remains unclear.

7. The paper uses clean benchmark datasets (VITON-HD, DressCode) with controlled lighting and backgrounds. It does not test PR-VTON on real-world data with noise (e.g., low-light images, cluttered backgrounds), which is common in practical VTON applications.

8. The method is trained and tested on the same datasets (VITON-HD, DressCode) without cross-dataset evaluation (e.g., training on VITON-HD and testing on a real dataset). This makes it difficult to assess its generality to unseen data distributions.

**Questions:**

- How does PRPE perform when adapted to DiT backbones other than FLUX.1-Fill-dev (e.g., Stable Diffusion 3, Flux.1-full)? Are modifications to RoPE required for different backbone architectures, or is PRPE truly plug-and-play?
- The loc-CMMD metric uses PP-OCRv5 with a confidence threshold of 0.8 to select text regions. What is the rationale for this threshold, and how would varying it (e.g., 0.7 or 0.9) affect the metric’s ability to distinguish high-quality from low-quality detail preservation?
- Could the PRL module be trained without ϕ_flow (i.e., fully self-supervised) by leveraging synthetic data or weak annotations? This would eliminate reliance on pre-trained flow models and improve end-to-end capability.
- How does PR-VTON handle garment occlusions (e.g., a hand holding a jacket, overlapping fabric)? The current method assumes clear garment-person correspondence, but occlusions would break this mapping—does PR-VTON have mechanisms to mitigate this?
- The paper claims PRPE improves "garment fit," but quantitative metrics for fit (e.g., shape alignment with body contours) are not reported. How could fit be objectively measured, and how does PR-VTON compare to SOTA in this aspect?

---

### Official Review · Reviewer_9sRK · 2025-10-29

**Soundness:** 2
**Presentation:** 2
**Contribution:** 2
**Rating:** 4
**Confidence:** 3

**Summary:**

This paper proposes PR-VTON, an enhanced virtual try-on method that improves fine-detail preservation by manipulating positional encodings (specifically RoPE) in Diffusion Transformer (DiT) architectures. The authors argue that by refining the positional encoding to reflect garment-to-body correspondences (via a mapping), they can guide attention mechanisms to better preserve high-frequency details like text logos and intricate patterns. The paper introduces: (1) Position-Refined Positional Encoding (PRPE) that remaps RoPE coordinates, (2) a Positional Relation Learning (PRL) module to learn the mapping φ, and (3) a new evaluation metric (loc-CMMD) focused on local texture fidelity. The method is evaluated on VITON-HD and DressCode benchmarks.

**Strengths:**

- The paper is tackling an interesting problem in the computer graphics and vision literature.
- The proposed method, in which positional encodings are manipulated to enhance both global coherence and fine detail rendering, is interesting and valuable. The zero-shot experiments shown in Figure 2 provide valuable intuitions onto why positional encoding matters.
- The paper aims to address real limitations in how existing evaluation metrics and protocols fail to capture fine semantic details which are relevant for virtual-try-on applications.
- The method shows consistent improvements across multiple metrics and datasets.
- The problem is clearly articulated and formulated and generally the paper is easy to follow.

**Weaknesses:**

- The paper consistently claims that their method requires "no additional modules", yet their main contribution is the inclusion of a module for better processing of positional-encoded information into the Diffusion Transformer. This contradition is strange and I urge the authors to either remove this claim or find a more accurate phrasing. The current writing really overstates contributions.
- The paper is missing critical technical details: There is not enough information of training times or computational cost. Further, there are missing details on the appearence flow predictor (are they re-training, is it frozen, fine-tuned?)
- How are layers 40, 41, 42 selected in the TransformerBlock? Is this an important hyperparameter?
- The covariance augmentation requires more information.
- Unproven claims: The paper claims to "markedly improve usability and scability", while insufficient empirical evidence is provided.
- The evaluation protocol requires further explanation and validation. It relies on completely arbitrary thresholdings and fallback strategies, as well as predefined image size and aspect ratio, all of which seem arbitrary and pose doubts on the validity of this evaluation protocol, therefore, this "contribution" is doubtful at best. There is no comparison with human perceptual studies.
- The flow-guided baseline claims to be "training-free", which is only partially true, as it requires a pretrained flow estimator, whose potential fallbacks are understudied.
- Maybe I misunderstood part of the paper but I think that notation is used in an inconsistent way throughout the paper.
- The paper is evaluated on FLUX-based DiT, which is an insufficient scope for proper contextualization in the literature.
- The paper claims that their method "integrates seamlessly with any DiT backbone" yet not sufficient evidence is provided. How does this module integrate in U-Net or SD based architectures?
- There is no discussion on failure cases of the method, nor any mention of problematic cases like occlusions, multiple garments, complex poses.
- Ablations are insufficient, only one hyperparameter is tested.
- The paper is not reproducible in its current state.
- There is little theoretical justification on the model design choices. Why does the RoPE modification work better than feature modification? Does PE really operate independently of the model? Does it not affect attention layers? What is the impact of this module in deeper of shallow layers?
- In general, the contribution is relatively small and constrained to relative improvements on the narrow field of virtual try-on, rendering its potential impact very limited.

**Questions:**

Other comments:
- Missing space between comma and "and" in line 57
- The "position-refined positional encoding" name could be improved, the word position appears twice and provides little extra information.

---

### Official Review · Reviewer_Q1EQ · 2025-10-31

**Soundness:** 1
**Presentation:** 3
**Contribution:** 1
**Rating:** 2
**Confidence:** 5

**Summary:**

The authors argue that treating the concatenated image as a single entity with default positional encoding (Fig. 2(1)) results in a generated image that completely fails to incorporate the input garment’s details. To address this, they propose introducing a coordinate mapping φ that assigns identical coordinates to semantically corresponding points on the garment and the target person, thereby reducing learning difficulty and better preserving fine-grained details. To achieve this, they employ a lightweight module (PRL) that distills a dense remapping field from an early warping-based try-on model, and they also introduce several new evaluation metrics.

**Strengths:**

1. The writing is clear and the motivation is easy to understand.

2. The comparison with warping and pasting in Table 2 is interesting.

**Weaknesses:**

1. On Motivation: As an experienced practitioner who has trained try-on models across multiple diffusion architectures—including SD 1.5, SDXL, SD3, and FLUX—I have observed that FLUX-fill inherently possesses a strong spatial understanding capability. Specifically, simply concatenating the garment image with an agnostic person image often yields correct try-on results with non-trivial probability. This suggests that FLUX-fill likely encountered such concatenated data during pretraining. Therefore, I question the core motivation illustrated in Fig. 2—namely, the claim that default positional encoding inevitably leads to failure.

2. On Motivation (continued): In my view, positional embeddings primarily serve to distinguish image dimensions and semantic categories. As long as the person image is consistently placed on the left and the garment on the right, there is no need to modify the positional encoding; the model can learn the semantic meaning of each spatial location through training alone. Alignment between person and garment should be handled within the attention mechanism—where person queries attend to corresponding garment keys, and the correct garment values are composited into the relevant regions. The alignment hinges on the compatibility of query and key representations, not on forcing positional embeddings to be identical. The authors’ approach of enforcing positional alignment is unnecessary and introduces several drawbacks, as detailed below.

3. On Theoretical Foundation: In the first training stage, the authors use Flow-warped positional encoding to adapt the DiT; in the second stage, they distill the Flow-warped model’s input using positional encoding injected at specific layers. This design raises three concerns:

First, the external optical-flow module is highly non-robust—it works reasonably well only on simple, front-facing poses with basic garment categories, but fails dramatically on complex poses (e.g., crossed arms). Diffusion-based try-on models, by contrast, generalize well across diverse scenarios. Injecting this fragile optical-flow mapping into FLUX may degrade its native spatial understanding and generative capability.

Second, the two-stage training pipeline is unnecessarily complex and difficult to implement or reproduce.
Third, regarding the PRL module: the semantic correspondence between latent features across different images and layers is not clearly defined. The authors manually select layers k∈{40,41,42} for mapping prediction based on empirical trials. This ad-hoc selection lacks theoretical grounding, is unstable, and feels inelegant. I doubt whether this choice leads to consistent or generalizable improvements.

4. On Experiments: The weight of the PE loss is set to
λ = 0.01, suggesting it is nearly negligible. More critically, the paper severely lacks qualitative visual comparisons, which are essential for evaluating try-on quality. Furthermore, the ablation study does not report results for the key components mentioned in the method—such as flow-warped PE, bbox PE, or learnable PE—but instead presents ablations on minor hyperparameters, significantly weakening the experimental validation.

5. On the Proposed Metrics: While I agree with the authors’ critique of existing evaluation metrics, the paper provides insufficient justification for the validity and reliability of the newly proposed metrics. Additionally, Table 2 omits results for the “warp-and-paste” baseline under the unpaired setting. To my understanding, warp-and-paste models tend to “cheat” in paired settings by memorizing alignments, but perform poorly on unpaired data—yet this crucial comparison is missing.

**Questions:**

Please refer to the weakness.

---

### Official Review · Reviewer_b3Nt · 2025-11-02

**Soundness:** 3
**Presentation:** 3
**Contribution:** 3
**Rating:** 6
**Confidence:** 4

**Summary:**

To preserve high-frequency details in virtual try-on, this paper propose PR-VTON (Position-Refined Virtual Try-On), a lightweight framework built on the Diffusion Transformer (DiT) backbone (FLUX.1-Fill-dev). PR-VTON optimizes positional encodings (PE) to steer attention toward semantically correct garment-body correspondences and propose PRL which is a light weight module that learns a stable remapping field. Additionally, the paper introduces a detail-focused evaluation metric (loc-CMMD).

**Strengths:**

1.The idea of finding the corresponding positional relationship between clothes and portraits and applying it to ROPE is innovative, and may bring inspiration to other tasks.

**Weaknesses:**

1.The introduction of PRL module implementation is not clear, especially Fig3.(3) is confusing.
2.There are few visualization results, and more comparison results with other methods are needed.

**Questions:**

1.In line 239, are the default coordinates of Cg and Cp reversed?
2.For cross-category try-on, such as replace a T-shirt to a dress, can the PRL module still correctly predict the corresponding relationships?

---

### Note · Authors · 2025-11-13

I have read and agree with the venue's withdrawal policy on behalf of myself and my co-authors.